# Association between epidemiological and clinico-pathological features of breast cancer with prognosis, family history, Ki-67 proliferation index and survival in Tunisian breast cancer patients

Najah Mighri[1], Nesrine Mejri[1,2], Maroua Boujemaa[1], Yosra Berrazega[2], Haifa Rachdi[2], Houda El Benna[1,2], Soumaya Labidi[1,2], Farouk Benna[3], Samir Boubaker[1,4], Hamouda Boussen[1,2], Sonia Abdelhak🄳[1], Yosr Hamdi🄳[1,4]*

1 Laboratory of Biomedical Genomics and Oncogenetics, LR20IPT05, Institut Pasteur de Tunis, University of Tunis El Manar, Tunis, Tunisia, 2 Medical Oncology Department, Abderrahman Mami Hospital, Faculty of Medicine Tunis, University Tunis El Manar, Tunis, Tunisia, 3 Department of Radiation Oncology, University of Tunis, Tunis, Tunisia, 4 Laboratory of Human and Experimental Pathology, Institut Pasteur de Tunis, Tunis, Tunisia

* yosr.hamdi@pasteur.tn

## Abstract

Breast cancer has different epidemio-clinical characteristics in Middle East and North-African populations compared to those reported in the Western countries. The aim of this study is to analyze the epidemiological and clinico-pathological features of breast cancer in Tunisia and to determine prognostic factors with special interest to family history, Ki-67 proliferation index and comorbidity. We retrospectively reviewed epidemiological and clinico-pathological data from patients' medical records, treated in the Medical Oncology Department at Abderrahmane Mami Hospital, in the period 2011–2015. Data has been collected on 602 breast cancer patients and analyzed using SPSS software V.23.0. Our study showed high fractions of young breast cancer patients and cases with dense breasts. The most prevalent comorbidities observed in the studied cohort were cardiovascular diseases and diabetes. Familial breast cancer was found in 23.3% of cases and was associated with younger age at diagnosis (p<0.001) and advanced stage (p = 0.015). Ki-67 index >20% was significantly associated with early age at diagnosis, lymph node involvement (p = 0.002), advanced tumor grade (p<0.001) and high risk of relapse (p = 0.007). Ki-67 cut-off 30% predicted survival in luminal cases. Survival was worse in patients with triple negative breast cancer compared to non-triple negative breast cancer, inflammatory breast cancer compared to non-inflammatory breast cancer, moderately to poorly differentiated tumors compared to well-differentiated tumors and with positive lymph nodes compared to pN0 (p<0.05). Our study showed new insights into epidemiological and clinico-pathological characteristics of breast cancer that are not well explored in Tunisian population. Considering our findings along with the implementation of electronic health record system may improve patient health care quality and disease management.

**Data Availability Statement:** All relevant data are within the manuscript and its Supporting information files.

**Funding:** This study was supported by the Tunisian Ministry of Health (PEC-4-TUN) and the Tunisian Ministry of Higher Education and Scientific Research (LR16IPT05 and LR20IPT05). The funders had no role in study design, data collection and analysis, decision to publish, or preparation of the manuscript.

**Competing interests:** The authors have declared that no competing interests exist.

## Introduction

Breast cancer is the most common malignancy among women and the first leading cause of cancer death in women worldwide, with 2.1 million new cases and 626.679 deaths predicted in 2018 [1]. In Tunisia, breast cancer represents 31.4% of all female malignancies with 2305 annual diagnosed cases in 2018 and with an incidence of 32.2 per 100.000 inhabitants making it a major public health concern [1]. Based on the latest World Health Statistics reports, breast cancer represents also the first cause of death from cancer among Tunisian females [2]. This heterogenous disease is characterized by several clinical and histological forms [3]. Indeed, large differences have been observed in the age of onset, stages at presentation, clinical manifestation and prognosis of breast cancer between various countries, mainly between Middle-East and North African breast cancer women and breast cancer patients from Western populations [4, 5].

Epidemio-clinical studies of breast cancer in Tunisia were mainly focused on breast cancer in young women [6–9], inflammatory breast cancer [10–12], prognostic factors for relapse [13, 14], survival [15–17] and the prognosis of molecular subtypes [18–20]. Indeed, Tunisian population was characterized by a younger age of onset of breast cancer when compared to Western countries and by a relatively high incidence rate among young patients under 35 years representing around 11% of all breast cancer cases in the country [21]. In addition, a high frequency of aggressive forms such as triple negative (TNBC) and inflammatory breast cancers (IBC) was also observed [5].

One of the strongest risk factors for breast cancer that is not well explored in Tunisian population is the presence of a positive family history of breast and ovarian cancers which has long been thought to indicate the presence of a strong inherited genetic component that predispose to the disease and to predict clinical characteristics and outcomes [22–25]. In addition, Ki-67 index which is a predictive and diagnostic biomarker that plays a significant prognostic role [26, 27], is not well studied in Tunisian breast cancer patients. Moreover, no Tunisian studies have studied comorbidities observed in breast cancer cohorts, although literature data showed that comorbidities negatively impacts overall breast cancer prognosis [28].

The aim of this report was to investigate epidemio-clinical characteristics of breast cancer in Tunisian population with special interest in clinico-pathological features and prognostic factors associated with familial breast cancer (FBC) cases. We also aimed to assess the clinico-pathological parameters associated with Ki-67 index value, comorbidities and survival among the studied cohort.

## Material and methods

### Patients

We retrospectively reviewed a cohort of 602 patients with histologically confirmed breast cancer treated in the period between 2011 and 2015, in Medical Oncology Department of "Abderrahmen Mami" Hospital which is the second national reference center for chemotherapy and radiation in Tunisia, with dedicated multidisciplinary teams. Data were collected from patient's medical record. The study was conducted according to the declaration of Helsinki and with the approval of the biomedical ethics committee of Institut Pasteur de Tunis (2017/16/E/Hôpital A-M). The need for consent has been waived by ethics committee, while ensuring additional data protection measures that preserve patient anonymity.

### Data collection

Several clinico-pathological parameters were investigated in this study including: age and delay at diagnosis (time between symptoms self-reported by patients and final histological

diagnosis of breast cancer), age at menarche, parity, age at first delivery, breastfeeding, oral contraceptive use, menopausal status, breast density, body mass index and comorbidities. Furthermore, consanguinity and data including information on family history of breast and ovarian cancers and other malignancies were also recorded. All patients had a biopsy with complete immunohistochemical evaluation before the initiation of any systemic therapy. TNM classification was based on reported clinical evaluation. In addition, clinico-pathological parameters have been collected including tumor size, lymph node involvement, Scarff-Bloom-Richardson (SBR) grade, hormone receptors and HER2 status, Ki-67 index, treatment type, relapse, outcome. Patients with metastatic disease were excluded.

Ki-67 (Clone MIB-1 (DAKO) dilution 1/100) was used for the automated immunohisto-chemical technique. The assessment procedure was manually performed on 10 fields using high magnification (x400). All the pathologists of the department were used to the estimation and evaluation of the ki-67 index. There was no centralized assessment and data was based on what was reported from multiple labs. In order to determine if there are special epidemiological and/or clinico-pathological differences between familial and sporadic breast cancer cases, we classified our cohort into two subgroups. The selection of familial cases was based on several criteria mainly the family history of breast and ovarian cancers and the age at diagnosis; patients were selected if at least one of the following criteria was fulfilled: (1) The patient was diagnosed with breast cancer before the age of 36 years, (2) The patient was diagnosed with triple negative breast cancer regardless of age, (3) The patient has at least two first or second-degree relatives with breast cancer, (4) The patient has at least two first or second-degree relatives with breast or ovarian cancer regardless of age, and at least one case of pancreatic or prostate cancer. Secondly, in order to study the correlation of Ki-67 index with the clinicopathological features and its survival prediction in the luminal breast cancer group, we used several Ki-67 cut-off points (14%, 20%, 30% and 50%). Authors evaluated these cut-offs based on previously published data [29]. Ki67 is already well studied in the literature as a continuous variable. Intrinsic subtype classification into Luminal A, Luminal B and Triple negative was based on immuno-histochemical criteria.

## Statistical analysis

Statistical analyses were performed using the Statistical Package for Social Sciences (SPSS Inc., Chicago, IL, USA) Version 23.0. Data were summarized by numbers and percentages for categorical variables, mean and range for continuous variables. Logistic regression was performed to evaluate prognostic factors.

The assessment of the association between familial breast cancer status, young age at onset, Ki-67 cut-off levels and prognostic factors was performed using Khi2 test or Fisher test. An independent t-test was used to compare the means of two studied groups.

We evaluated the prognostic value of several cut-off levels of Ki-67 in terms of overall survival (OS). We also considered different subgroups according to axillary lymph node involvement: pN0, 1-3pN+ and ≥4pN+. Survival curves were dressed according to the Kaplan-Meier analysis and compared with the log-rank test. All $p$-values were two-sided, and $p < 0.05$ was used to indicate a statistically significant difference.

## Results

A total of 602 patients with a histologically proven breast cancer have been included in this study. Epidemiological features and clinico-pathological characteristics of the studied population were presented in S1 and S2 Tables. The collection of these epidemiological and clinico-pathological parameters allowed us to set up an electronic database in the Oncology

Department of Abderrahmane Mami hospital, which facilitates the storage, retrieval modification, the management of patients medical records and their follow-up.

## Epidemiological features

Among breast cancer patients that participated in the current study, 54.6% were premenopausal. At the time of their breast cancer diagnosis, 4.8% of patients were nulliparous and 2.9% were pregnant. Mean age at menarche was 12.9±1.618 years. Mean age at first childbirth was 25.9±10.727 years. 72.15% have received oral contraception and 22.6% of patients have never breastfed. According to the Breast Imaging-reporting and Data System (BI-RADS) classification, 56% of patients had dense breasts. Obesity defined by BMI $\geq$30kg/m$^2$ was seen in 28% of cases. We considered comorbidities known to be associated with breast cancer from the literature such as diabetes, hypertension, dyslipidemia, metabolic syndrome and rheumatologic diseases. Most prevalent comorbidities observed in our breast cancer cohort were cardiovascular diseases (50.78%) and diabetes (18.32%). 25.3% of patients were from consanguineous families. Familial cases represented 23.3% of breast cancer patients. The family history of ovarian cancer was observed in as few as 2% of patients. In addition, personal history of cancer was found in 2% of the studied patients (S1 Table). All patients were Tunisian without other ethnicity groups presented.

## Clinicopathological features

The mean age at diagnosis of breast cancer was 48.7±11.417 years (ranging from 22 to 83 years) and 13.4% of patients were $\leq$35 years. The mean diagnostic delay was 8 months and 39% of patients reported a delay in diagnosis of 6 months or less before their first consultation after the onset of symptoms. Mean clinical tumor size at diagnosis was 38.95 mm. Locally advanced tumors (T3, T4) were seen in 21.6% of patients. Data for histological tumor type showed that invasive ductal carcinoma (IDC) was the most frequent (91.4%) while infiltrating lobular carcinoma (ILC) was observed in 2.7% of cases. The presence of the intraductal component was observed in 50% of breast cancer patients. Multifocality defined as the presence of 2 or more separate invasive tumors was noticed in 29.7% of cases. Upper outer and upper inner quadrants were mostly affected in 47.3% and 13.1% of patients, respectively. SBR grade II was the most frequent (53.1%). Approximately, 50% of patients were diagnosed with lymph node positive disease. The median Ki-67 value was 30% (range, 1–90%). Luminal B was the most common subtype (46.26%) followed by luminal A (27.95%), TNBC (15.47%), and HER2 + (10.32%). Luminal B patients with HER2 negative disease represented 35.9% (S2 Table). Distant metastases at diagnosis were observed in 14.5% of patients. Among non-metastatic cases, 21.7% have relapsed and the most frequent sites of relapses were bones and lungs.

## Treatment of breast cancer cases

Nearly 34.5% of patients had tumorectomy and 65.5% had mastectomy. Neoadjuvant chemotherapy (CT) was performed in 124 cases (14%) and adjuvant chemotherapy was performed in 406 patients (67.9%). From 406 patients who received adjuvant chemotherapy, 83.6% had sequential Epirubicin and Taxane-based chemotherapy. 83.4% received adjuvant radiation therapy with a mean dose of 55 Gray. Three hundred and sixty-two patients (67.9%) had adjuvant hormonal therapy: 212 started with tamoxifen and 149 with aromatase-inhibitors. From patients whose tumors overexpressed the HER2 biomarker, 84.2% received Trastuzumab. The following part of our report will be devoted to the study of particular epidemio-clinical parameters mainly family history, early onset disease ($\leq$35 years), Ki-67 proliferation index, comorbidity and survival.

## Comparison of epidemiological characteristics and clinico-pathological features between familial and sporadic breast cancer patients

Familial breast cancer has been observed in 139 out of 596 patients (23.3%) (Table 1). There was a significant difference in the distribution of age at diagnosis among familial and sporadic breast cancer cases (mean 42.46 vs 50.63 years) (Table 1). Indeed, familial breast cancer cases were significantly younger than sporadic patients (p<0.001). Furthermore, familial breast cancer patients were more likely to be premenopausal (p = 0.001). A delay in diagnosis was observed in sporadic cases (9.47 months) when compared to familial cases (6.19 months) but no significant difference between the two groups was observed. No significant differences have been also observed between the two groups in histological subtype, nodal status, SBR grade, hormonal receptors status, HER2 receptor expression, Ki-67 index, metastases, and relapse status. However, a statistically significant difference was observed regarding intraductal component (p = 0.027) and a significantly high tumor stage was noted among familial cases (p = 0.048) (Table 1).

Overall survival did not differ between non-metastatic familial and sporadic cases (p = 0.242) (Fig 1).

## Clinico-pathological features of breast cancer in young patients aged ≤35 years

Our results showed, early onset breast cancer patients (≤35 years) accounted for 13.4% of the total breast cancer cases. A statistically significant difference in diagnostic delay between young (≤35 years) and older (>35 years) patients was observed (p = 0.038). Older patients had longer delays in diagnosis than young patients (more than 6 months). However, no significant differences have been observed between the two age groups in molecular subtypes, hormonal receptors status, HER2 status, nodal involvement, SBR grade, Ki-67 index, metastases and relapse (Table 2).

## Correlation between Ki-67 proliferation index and clinicopathological features of breast cancer and its role in predicting survival in luminal breast cancer

In order to evaluate the value of Ki-67 as a prognostic marker in Tunisian breast cancer patients and to analyze its association with several clinicopathological parameters, we used different Ki-67 cut-off points. Ki-67-high tumors (>20%) were significantly associated with early onset age of breast cancer (p = 0.002), lymph node involvement (p = 0.002) (pN0 vs pN+), high SBR grade (p<0.001) (Grade I vs Grade II-II), ER negativity (p = 0.001), PR negativity (p<0.001), and with positive HER2 receptor (p = 0.001). Furthermore, breast cancer patients with Ki-67 higher than 20% exhibited higher incidence of relapse than those with Ki-67 less than 20% (p = 0.007). Moreover, a statistically significant difference was observed between Ki-67>20% and delayed diagnosis for up to 6 months (p = 0.004) (Table 3).

Ki-67 expression is used for subdividing luminal breast cancers into luminal A and luminal B groups which tend to grow slowly and have a good prognosis compared to non-luminal breast cancer. In order to evaluate the cut-off value of Ki-67 that predicted survival in the luminal breast cancer group and to investigate its survival impact according to axillary lymph node involvement, we retrospectively selected early breast cancer cases with HR positive tumours. Median Ki-67 value was 28%. Median follow-up was 38 months. In the overall population of luminal breast cancer, we only observed a significant difference in OS with the Ki-67 cut-off of 30% (67 vs 64 months, p = 0.04, HR = 2.76 95% CI [0.89–3.73]). In node negative and in 1-

**Table 1. Comparison of epidemiological and clinicopathological parameters between familial and sporadic breast cancer patients.**

| Variables | Familial cases | Sporadic cases | *P* value* |
|---|---|---|---|
| | N = 139 | N = 457 | |
| Mean age at diagnosis (years) | 42.46±11.951 | 50.63±10.587 | **<0.001** |
| Mean diagnostic delay (months) | 6.19 | 9.47 | 0.187 |
| **Menopausal status** | | | |
| Premenopausal | 93/138 (67.39) | 228/451 (50.55) | **0.001** |
| Postmenopausal | 45/138 (32.61) | 223/451 (49.45) | |
| **Breast density** | | | |
| Dense breasts | 15/28 (53.57) | 45/79 (56.96) | 0.756 |
| Non-dense breasts | 13/28 (46.43) | 34/79 (54.43) | |
| **Histological type** | | | |
| IDC | 126/139 (90.65) | 418/457 (91.47) | 0.896 |
| ILC | 5/139 (3.6) | 11/457 (2.40) | |
| MC | 5/139 (3.6) | 17/457 (3.72) | |
| Other | 3/139 (2.15) | 11/457 (2.41) | |
| **Histological grade status** | | | |
| Grade I | 13/134 (9.7) | 49/444 (11.04) | 0.889 |
| Grade II | 71/134 (52.99) | 236/444 (53.15) | |
| Grade III | 50/134 (37.31) | 159/444 (35.81) | |
| **T stage** | | | |
| T1-T2 | 82/112 (73.21) | 311/399 (77.94) | **0.048** |
| T3 | 16/112 (14.29) | 28/399 (7.02) | |
| T4 | 14/112 (12.5) | 60/399 (15.04) | |
| **Intraductal component** | | | |
| Yes | 40/99 (40.40) | 185/349 (53) | **0.027** |
| No | 59/99 (59.6) | 164/349 (47) | |
| **Nodes involvement** | | | |
| N+ | 63/118 (53.39) | 230/388 (59.28) | 0.257 |
| N- | 55/118 (46.61) | 158/388 (40.72) | |
| **Ki-67 index status** | | | |
| Ki-67 ≤20% | 43/105 (40.95) | 169/398 (42.46) | 0.780 |
| Ki-67>20% | 62/105 (59.05) | 229/398 (57.54) | |
| **Molecular subtypes** | | | |
| Luminal A | 40/138 (28.99) | 126/457 (27.57) | 0.247 |
| Luminal B | 68/138 (49.27) | 208/457 (45.51) | |
| HER2+ | 8/138 (5.80) | 54/457 (11.82) | |
| TNBC | 22/138 (15.94) | 69/457 (15.10) | |
| **ER status** | | | |
| ER+ | 105/137 (76.64) | 327/457 (71.55) | 0.241 |
| ER- | 32/137 (23.36) | 130/457 (28.45) | |
| **PR status** | | | |
| PR+ | 93/137 (67.88) | 289/457 (63.24) | 0.320 |
| PR- | 44/137 (32.12) | 168/457 (36.76) | |
| **HER2 status** | | | |
| HER2+ | 42/138 (30.43) | 151/456 | 0.556 |
| HER2- | 96/138 | 305/456 | |
| **Metastatic status** | | | |

(*Continued*)

**Table 1.** (Continued)

| Variables | Familial cases | Sporadic cases | *P* value* |
|---|---|---|---|
| | N = 139 | N = 457 | |
| M0 | 118/138 (85.51) | 390/456 (85.53) | 0.996 |
| M1 | 20/138 (14.49) | 66/456 (14.47) | |
| Relapse | | | |
| Yes | 38/131 (29) | 89/401 (22.19) | 0.112 |
| No | 93/131 (71) | 312/401 (77.81) | |

* Fisher's exact test is used instead of Khi2 test if one or more variables had an expected frequency of less than five.

**Abbreviations:** IDC = Invasive Ductal Carcinoma; ILC = Invasive Lobular Carcinoma; MC = Mixed carcinoma; ER = Estrogen Receptor; PR = Progesterone Receptor; HER2 = Human Epidermal Growth Factor Receptor2; TNBC = Triple Negative Breast Cancer.

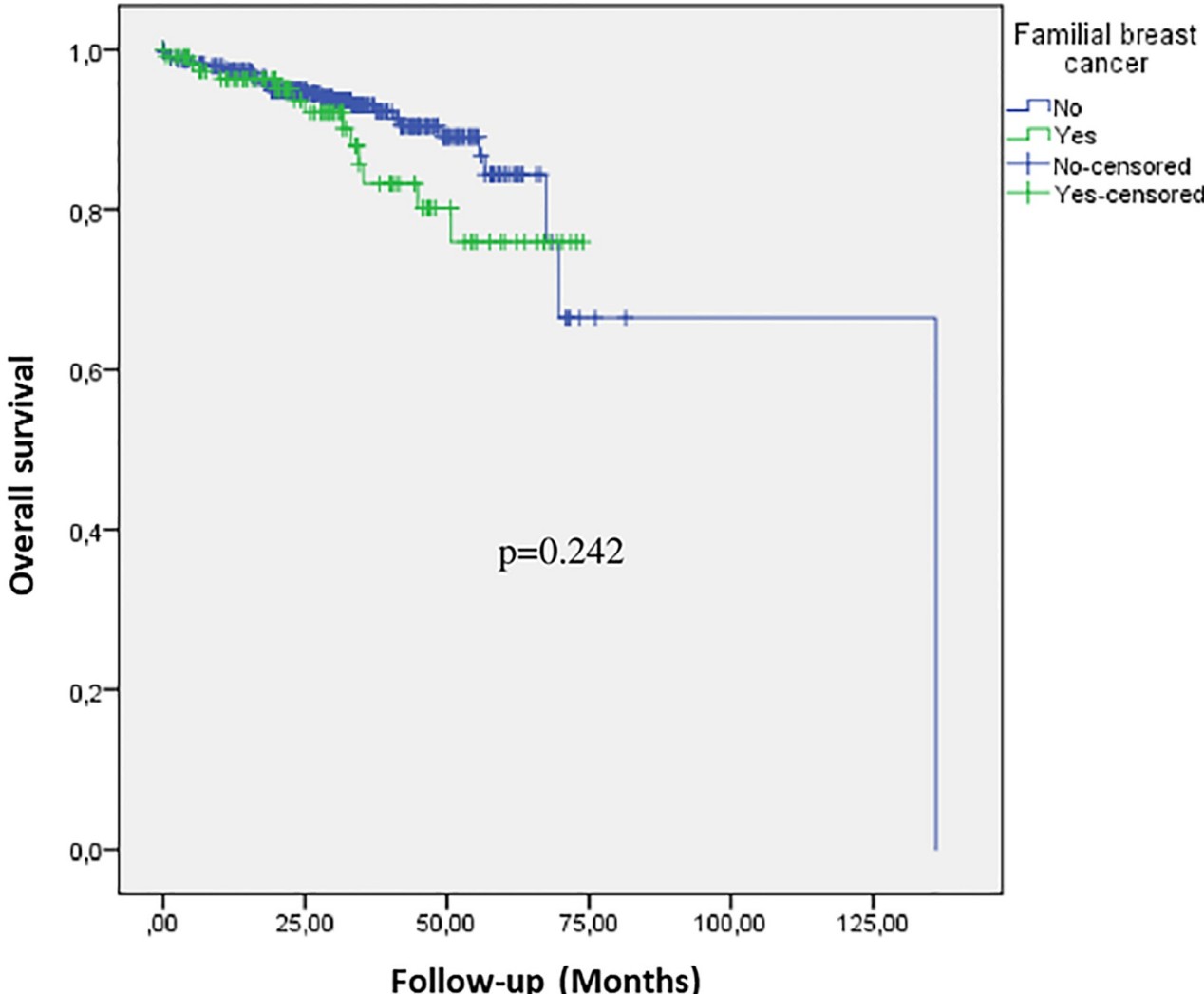

**Fig 1. Kaplan Meier estimates overall survival according to familial breast cancer status.** Blue line represents sporadic and green line familial cases.

**Table 2. Comparison of clinico-pathological features between 2 age groups (breast cancer patients ≤35 years and >35 years).**

| Variables | BC patients (≤35 years) | BC patients (>35 years) | P value* |
|---|---|---|---|
| | N = 80 | N = 516 | |
| **Diagnostic delay (6months)** | | | |
| **Yes** | 15/56 (26.79) | 133/321 (41.43) | **0.038** |
| **No** | 41/56 (73.21) | 188/321(58.57) | |
| **Histological grade status** | | | |
| **Grade I** | 5/76 (6.58) | 57/502 (11.35) | 0.182 |
| **Grade II** | 37/76 (48.68) | 269/502 (53.59) | |
| **Grade III** | 34/76 (44.74) | 176/502 (35.06) | |
| **Intraductal component** | | | |
| **Yes** | 22/56 (39.29) | 203/393 (51.65) | 0.083 |
| **No** | 34/56 (60.71) | 190/393 (48.35) | |
| **Nodes involvement** | | | |
| **N+** | 35/69 (50.72) | 259/438 (59.13) | 0.188 |
| **N-** | 34/69 (49.28) | 179/438 (40.87) | |
| **Molecular subtypes** | | | |
| **Luminal A** | 24/79 (30.38) | 142/516 (27.52) | 0.619 |
| **Luminal B** | 34/79 (43.04) | 242/516 (46.90) | |
| **HER2+** | 6/79 (7.59) | 55/516 (10.66) | |
| **TNBC** | 15/79 (18.99) | 77/516 (14.92) | |
| **ER status** | | | |
| **ER+** | 55/78 (70.51) | 377/516 (73.06) | 0.638 |
| **ER-** | 23/78 (29.49) | 139/516 (26.94) | |
| **PR status** | | | |
| **PR+** | 50/78 (64.10) | 332/516 (64.34) | 0.967 |
| **PR-** | 28/78 (35.90) | 184/516 (35.66) | |
| **HER2 status** | | | |
| **HER2+** | 23/79 (29.11) | 168/515 (32.62) | 0.534 |
| **HER2-** | 56/79 (70.89) | 347/515 (67.38) | |
| **Metastatic status** | | | |
| **M0** | 69/79 (87.34) | 439/515 (85.24) | 0.622 |
| **M1** | 10/79 (12.66) | 76/515 (14.76) | |
| **Relapse** | | | |
| **Yes** | 23/73 (31.51) | 106/458 (23.14) | 0.122 |
| **No** | 50/73 (68.49) | 352/458 (76.86) | |

* Fisher's exact test is used instead of Khi2 test if one or more variables had an expected frequency of less than five.

**Abbreviations:** ER = Estrogen Receptor; PR = Progesterone Receptor; HER2 = Human Epidermal Growth Factor Receptor2; TNBC = Triple Negative Breast Cancer.

3pN+ positive tumors, there was no significant impact on the survival for different Ki-67 cut-off values. In ≥4pN+ group, where patients with Ki-67>50% had significantly worse OS compared to patients ≤50% (63 vs 30 months, p = 0.01, HR = 11.6 95% CI [1.5–16]) (Table 4).

## Comorbidities and breast cancer

The most prevalent comorbidities observed in our breast cancer cohort were cardiovascular diseases (50.78%) and diabetes (18.32%). This study shows that women older than 35 years at

**Table 3. Correlation between Ki-67 expression and clinicopathological features in breast cancer patients.**

| Variables | Ki-67 ≤20% | Ki-67 >20% | P value* |
|---|---|---|---|
| | N = 216 | N = 293 | |
| **Early age at onset** | | | |
| ≤35 years | 12/216 (5.56) | 41/293 (14) | **0.002** |
| >35 years | 204/216 (94.44) | 252/293 (86) | |
| **Delayed diagnosis (6 months)** | | | |
| Yes | 67/142 (47.18) | 57/182 (31.32) | **0.004** |
| No | 75/142 (52.82) | 125/182 (68.68) | |
| **Histological type** | | | |
| IDC | 193/216 (89.35) | 269/293 (91.81) | 0.694 |
| ILC | 6/216 (2.78) | 7/293 (2.39) | |
| MC | 11/216 (5.09) | 9/293 (3.07) | |
| Other | 6/216 (2.78) | 8/293 (2.73) | |
| **Histological grade status** | | | |
| I | 36/213 (16.90) | 16/286 (5.59) | **<0.001** |
| II- III | 177/213 (83.1) | 270/286 (94.41) | |
| **Intraductal component** | | | |
| Yes | 93/178 (52.25) | 108/218 (49.54) | 0.592 |
| No | 85/178 (47.75) | 110/218 (50.46) | |
| **Lymph node involvement** | | | |
| N+ | 97/197 (49.24) | 153/238 (64.29) | **0.002** |
| N- | 100/197 (50.76) | 85/238 (35.71) | |
| **Massive lymph node involvement** | | | |
| >3N+ | 33/97 (34.02) | 80/153 (52.29) | **0.005** |
| <3N+ | 64/97 (65.98) | 73/153 (47.71) | |
| **ER status** | | | |
| ER+ | 174/216 (80.55) | 196/293 (66.89) | **0.001** |
| ER- | 42/216 (19.44) | 97/293 (33.11) | |
| **PR status** | | | |
| PR+ | 160/216 (74.07) | 173/293 (59.04) | **<0.001** |
| PR- | 56/216 (25.93) | 120/293 (40.96) | |
| **HER2 status** | | | |
| HER2+ | 51/215 (23.72) | 110/293 (37.54) | **0.001** |
| HER2- | 164/215 (76.28) | 183/293 (62.46) | |
| **Metastatic status** | | | |
| M0 | 198/215 (92.09) | 243/293 (82.94) | **0.003** |
| M1 | 17/215 (7.91) | 50/293 (17.06) | |
| **Relapse** | | | |
| Yes | 32/204 (15.69) | 67/257 (26.07) | **0.007** |
| No | 172/204 (84.31) | 190/257 (73.93) | |

* Khi2 test was conducted.

**Abbreviations:** IDC = Invasive Ductal Carcinoma; ILC = Invasive Lobular Carcinoma; MC = Mixed carcinoma; ER = Estrogen Receptor; PR = Progesterone Receptor; HER2 = Human Epidermal Growth Factor Receptor2.

**Table 4. Ki-67 cut-offs predicting survival in overall luminal breast cancer tumours.**

| Ki-67 cut-off | All luminal breast cancer group (M median survival) | | | N0 | | | 1–3 pN+ | | | ≥4 p N+ | | |
|---|---|---|---|---|---|---|---|---|---|---|---|---|
| | M | p | HR [CI] | M | p | HR [CI] | M | p | HR [CI] | M | p | HR [CI] |
| ≤14% | 68 | 0.33 | 0,41 [0.09–0.82] | 69 | 0.89 | 1.18 [0.1–13] | 61 | 0.45 | 0.2 [0.3–9.2] | 64 | 0.19 | 0.35 [0.1–12] |
| >14% | 67 | | | 64 | | | 59 | | | 59 | | |
| ≤20% | 67 | 0.36 | 1.63 [0.57–4.63] | 70 | 0.77 | 0.7 [0.6–7.8] | 65 | 0.26 | 3.27 [0.3–31] | 67 | 0.44 | 2.46 [0.2–22] |
| >20% | 66 | | | 58 | | | 52 | | | 56 | | |
| ≤30% | 67 | 0.04 | 2.76 [0.89–3.73] | 69 | 0.8 | 1.36 [0.2–15] | 65 | 0.41 | 2.27 [0.3–16] | 63 | 0.05 | 6.8 [0.7–9.3] |
| >30% | 64 | | | 59 | | | 52 | | | 55 | | |
| ≤50% | 65 | 0.21 | 2.07 [0.66–6.53] | 56 | 0.65 | 1.97 [0.7–4.7] | 61 | 0.97 | 1.36 [0.4–13] | 63 | 0.01 | 11.6 [1.5–16] |
| >50% | 65 | | | 53 | | | 54 | | | 30 | | |

**Abbreviations:** M = Median; HR = Hazard Ratio, CI = Confidence Interval.

Survival data were dressed according to the Kaplan-Meier analysis and compared with the log-rank test.

the time of their breast cancer diagnosis, had more prevalence of comorbid conditions than younger breast cancer patients (p<0.001). However, no association has been identified between comorbidities and relapse among breast cancer patients (p = 0.075). Similarly, comorbidity was not significantly associated with overall survival in breast cancer patients (p = 0.079) (Fig 2).

### Survival in non-metastatic breast cancer patients

Several clinico-pathological factors that could affect survival of women diagnosed with non-metastatic breast cancer have been assessed such as: grade, stage, molecular subtypes, HR status, HER2 status, Ki-67 expression, and nodes involvement.

Our findings showed that overall survival was worse in patients with TNBC (p<0.001) compared to non-TNBC, IBC (p = 0.024) compared to non-IBC, moderately to poorly differentiated tumors (p = 0.013) compared to well-differentiated tumors and with lymph node involvement (p = 0.018) compared to pN0.

## Discussion

Breast cancer is a major public health challenge in low and middle income countries (LMICs). In this paper, we described work in progress to develop the first breast cancer database in Tunisia. In fact, the development of this database represents a first step towards the implementation of electronic medical records systems to support safe and standardized cancer care in low-resource settings. In addition, to ensure collection of data relevant to breast cancer phenotypes, a standardized data collection instrument was developed in order to facilitate the harmonization of data which have not been collected in a harmonized way. These outcomes will help improving the quality and efficiency of cancer care and facilitate communication between healthcare personnel, researchers, and patients. This database includes clinico-pathological features of 602 Tunisian breast cancer cases that have been investigated in the current study. The mean age at diagnosis of breast cancer patients was 48.7 years. This result is similar to those described in previous studies on breast cancer in Arab women and different from that reported in patients from Western countries [30, 31]. Inflammatory and triple-negative breast tumors were found among 5.35% and 15.5% of patients, respectively. These proportions of aggressive breast cancer forms are different from those described in previous Tunisian studies

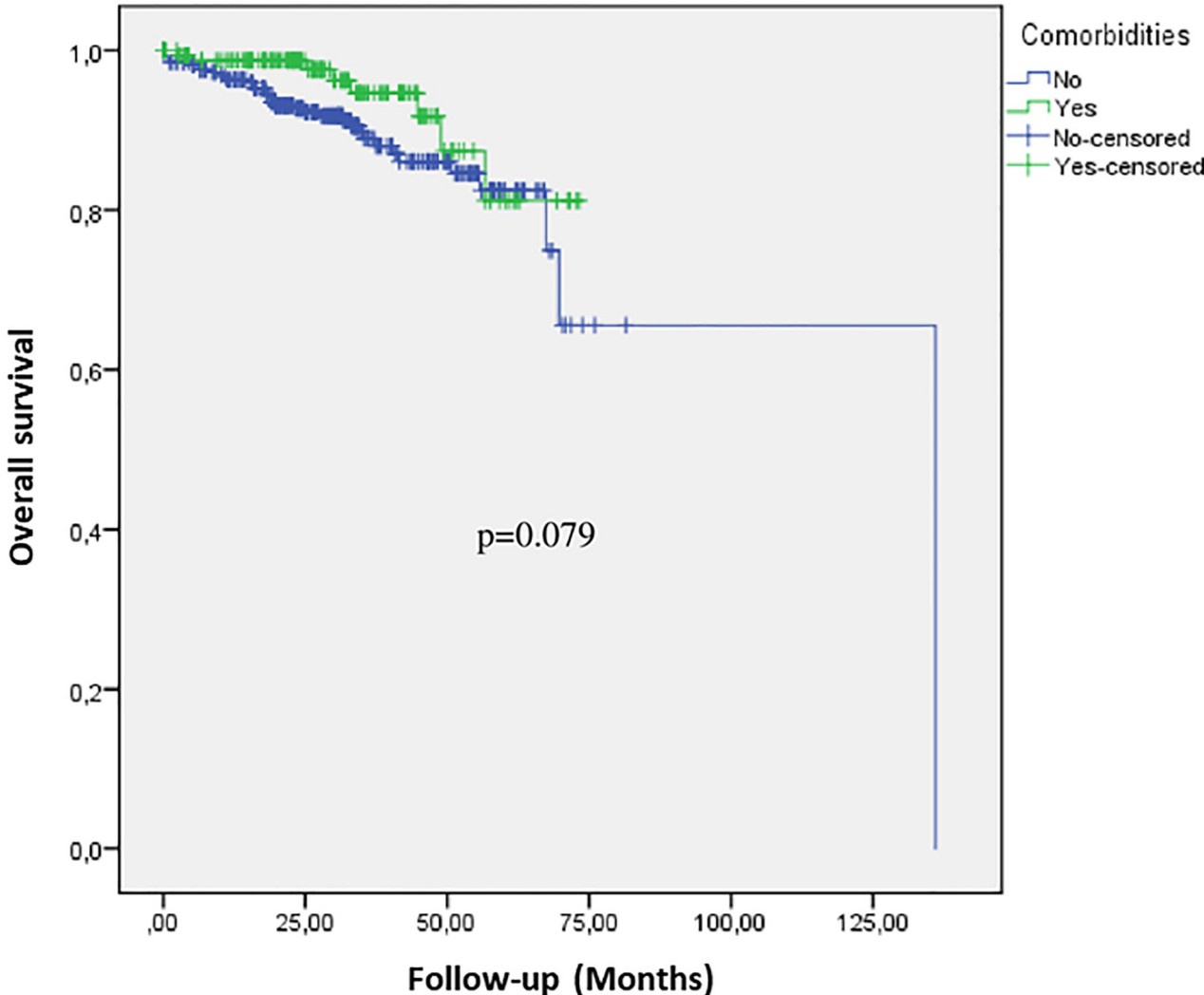

**Fig 2. Kaplan Meier estimates overall survival according to comorbidity status.** Blue line represents non-comorbid patients and green line comorbid cases.

(TNBC = 22.5%, IBC = 7–10%) [12, 18, 32] and could be considered as an update of latest data on these breast cancer forms in Tunisian population. Furthermore, our results were different to those described in several other populations, such as Western and Sub-Saharan Africa where TNBC accounts for 27% to 61% of cases. For IBC, it represents 2% of breast cancer in Europe and USA, lower than that reported in North Africa and Tunisia (5–10%) [33, 34]. Furthermore, a delayed diagnosis for up to 6 months after first detection of a breast mass has been observed among 39% of patients. Large mean tumor size, high proportions of T3 and T4 stages, lymph nodes involvement and patients diagnosed with distant metastases were observed. All these characteristics are considered as factors associated with late diagnosis of breast cancer. These results showed that despite large awareness campaigns running in Tunisia, breast cancer is still shrouded in secrecy and is still considered a taboo for several socio-cultural reasons. In our study, luminal B cancer was more prevalent (46.27%) than luminal A (28%). This result is in discordance with previous Tunisian studies [18, 19] which could be explained by the heterogeneity of breast cancer even in different cohorts drawn from the same

country. The Inclusion of HER2 positive cases in luminal B groups could also be an explanation. In addition, our results are in discordance with previous studies conducted in several countries all over the world including Algeria [35], Egypt [36], Japan [37] and the USA [38]. Although, other studies conducted in Saudi Arabia and Italy found luminal B subtype more common than luminal A [39, 40]. This variation in the distribution of breast cancer molecular subtypes could be explained by the fact that different cut offs value of Ki-67 index are used for the stratification of luminal breast cancer groups across the world which may lead to mis-classification bias. In Tunisian population, the Ki-67 index which is one of the most controversially discussed breast cancer biomarkers [41, 42], was not well investigated. Therefore, we evaluated its clinical significance as a prognostic marker and we analyzed its association with clinico-pathological parameters in breast cancer patients as well as its role in predicting survival among the luminal breast cancer group. Our results demonstrated that there was a statistically significant difference between the 2 groups (Ki-67≤20%) and (Ki-67>20%) regarding a set of parameters. Tumor with Ki-67>20% showed the poorest prognosis; early age at onset, advanced tumors grade, positive node involvement, high risk of relapse (p<0.05). In accordance with our results, other studies found that a higher Ki-67 index is significantly correlated with positive lymph nodes [43, 44], an increased risk of recurrence [45], a high grading [43] and an early age at onset [46]. In our study, higher Ki-67 expression (>20%) was also more frequently associated with HR-negative and HER2-positive tumors. This result was in agreement with studies who reported that a higher Ki-67 index significantly correlated with HER2-positive breast tumors [41, 43] and negatively correlated with HR positivity [47].

Additional analyses performed in the present study demonstrated that Ki-67 index predicted survival with a cut-off value of 30% in the overall luminal breast cancer group and 50% in ≥4pN+ studied tumors. Previous report demonstrated that a Ki-67 index with a cut-off (≥20%) is significantly correlated with poorer prognosis and early recurrence, particularly in luminal A type tumors [43]. Therefore, our study is first to evaluate Ki67 association with clinico-pathological features and outcome among Tunisian breast cancer patients and significant associations were observed. According to these results high levels above 30% could be used to classify tumors into Luminal B, but the data does not confirm that a Ki67 below 30% can be interpreted as luminal A. Furthermore, due to the lack of external validation study, caution should be used when implementing this cut-off as an independent marker; all other clinico-pathological parameters should also be considered when evaluating the prognosis of any given patient.

Tumour location was higher in the upper outer quadrant (UOQ) (47.3%). Indeed, tumour location is highest in the UOQ (50–58%) across multiple populations [48] and the high proportion of UOQ carcinomas of the breasts reflects the greater amount of breast tissue in this quadrant [49].

Apart from the characteristics of breast cancer described above, several risk factors such as coexistence of chronic diseases at the time of breast cancer diagnosis are reported. This comorbidity potentially affects the stage at diagnosis, treatment, and outcomes of cancer patients [50]. In our study, no association has been revealed between relapse, survival and comorbidity and most of the comorbid conditions are chronic diseases such as cardiovascular diseases, diabetes and metabolic syndrome which are partly associated with a non-healthy lifestyle behavior. These results can be considered as a form of bias and could be explained by the fact that oncologists usually ask for these particular diseases while they do not ask about other hereditary conditions related to breast cancer, such as Li-Fraumeni, Fanconi anemia, Ataxia telangiectasia and Lynch syndrome which may have an effect on the diagnosis, treatment decision, disease evolution and family healthcare which may explain the few fraction of patients with family history of ovarian cancer that could be underestimated in the studied cohort. Moreover,

family history of breast cancer is an established risk factor for the development of the disease [22, 51]. In our study, familial cases represented 23.3% of all breast cancer patients. We based our classification on anamnestic and clinical parameters as well as tumour subtype not on genetic testing. Nevertheless, about 5–10% of all breast cancer cases are thought to be familial. The high fraction of FBC identified in our study could be explained by the special and specific genetic architecture of Tunisian population such as the high consanguinity and endogamy rates. Our results also showed that familial breast cancer seems to affect young and premeno-pausal women. Similarly, other reports noted a significantly higher frequency of premeno-pausal women among FBC patients [52, 53] and demonstrated that women with family history were more likely to be diagnosed at an earlier age [54, 55]. Several studies have revealed that FBC has some specific clinical features such as bilateral breast cancer, advanced stage, lymph node involvement and negative hormone receptors compared to sporadic cases [56, 57]. Our study has shown that FBC cases were significantly younger than sporadic patients and were diagnosed at later stages (T3). Rising awareness about the importance of the *BRCA1/2* genetic testing among Tunisian patients with a strong family history of breast and/or ovarian cancer will eventually lead to early disease detection. In our study, 13.4% of patients were ≤35 years. This fraction is higher than that reported previously in Tunisia (11%) [21] and this is a brand new update of breast cancer in young Tunisian patients. Compared to older patients, young women generally face more aggressive cancers [58, 59]. In our study, we observed that patients over the age of 35 had longer delays in diagnosis when compared to young patients. Although most breast cancers occur at older ages, young patients may be more prone to screening proce-dures and self-examination due to their implication in awareness campaigns.

In the current study, we highlighted the epidemio-clinical specificities of breast cancer among Tunisian patients with special focus on parameters that were not well investigated in previous Tunisian studies. However, other crucial parameters were not explored in our study due to some missing data such as geographical origin, place of residence, socio-economic and educational level. Results on other parameters like breast density were not conclusive due to the lack of precise scores generated from automated technique for measurement of volumetric mammographic density that were absent in our country.

In this study, we propose to use a new form model (S1 File) by the oncologists during the first consultation which contains more details regarding the family history of cancers, consan-guinity and comorbidities with special interest to rare hereditary diseases.

To summarize, to the best of our knowledge, the present study is the first to assess the asso-ciation of familial breast cancer, and Ki-67 index with clinico-pathological characteristics and prognostic factors of breast cancer in Tunisian population. We demonstrated that clinico-pathological profile of familial cases differs from sporadic patients. We also showed that Ki-67 cut-off >20% may be considered as a valuable biomarker in Tunisian breast cancer patients and can be used in their follow-up. This biomarker predicted survival with a cut-offs value of 30% in luminal breast cancer patients and 50% in this same group with more than 4pN+. These cut-offs should be more investigated to select patients for adjuvant therapy. The identifi-cation of clinico-pathological specificities and prognostic factors of breast cancer in Tunisia will contribute to a better management of the disease and for better health outcomes.

## Conclusions

Our study investigated clinico-pathological features of breast cancer that were not well studied in Tunisian population. Our findings show that despite the influence of awareness efforts con-ducted in Tunisia and advanced screening and detection techniques, the disease remains undetected until it reaches advanced stages. Based on our results, a multidisciplinary approach

needs to be employed with implementation of electronic health records systems in order to promote health care coordination and communication. We further recommend more breast cancer awareness programs targeting all medical oncology stakeholders such as radiologists, pathologists and geneticists.

## Supporting information

**S1 Table. Epidemiological features of breast cancer patients.**
(DOCX)

**S2 Table. Clinicopathological characteristics of breast cancer patients.**
(DOCX)

**S1 File. Form model proposed for the investigation of epidemiological and genetics features of breast cancer in Tunisia.**
(DOCX)

## Acknowledgments

We would like to thank all patients, researchers, clinicians and administrative staff of the Medical Oncology Department of Abderrahmen Mami Hospital who have enabled this work to be carried out.

## Author Contributions

**Conceptualization:** Najah Mighri, Nesrine Mejri, Yosr Hamdi.

**Data curation:** Najah Mighri, Nesrine Mejri, Maroua Boujemaa, Yosra Berrazega, Haifa Rachdi, Houda El Benna, Soumaya Labidi.

**Formal analysis:** Najah Mighri, Nesrine Mejri, Yosr Hamdi.

**Investigation:** Najah Mighri, Maroua Boujemaa, Yosra Berrazega, Haifa Rachdi, Soumaya Labidi.

**Methodology:** Najah Mighri, Nesrine Mejri, Yosra Berrazega, Haifa Rachdi, Houda El Benna, Soumaya Labidi, Yosr Hamdi.

**Software:** Najah Mighri, Nesrine Mejri, Maroua Boujemaa, Yosra Berrazega, Haifa Rachdi, Houda El Benna, Soumaya Labidi, Yosr Hamdi.

**Supervision:** Nesrine Mejri, Samir Boubaker, Hamouda Boussen, Sonia Abdelhak.

**Validation:** Nesrine Mejri, Farouk Benna, Samir Boubaker, Hamouda Boussen, Sonia Abdelhak, Yosr Hamdi.

**Writing – original draft:** Najah Mighri, Houda El Benna.

**Writing – review & editing:** Nesrine Mejri, Maroua Boujemaa, Yosra Berrazega, Haifa Rachdi, Soumaya Labidi, Farouk Benna, Samir Boubaker, Hamouda Boussen, Sonia Abdelhak, Yosr Hamdi.

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
