## [Decision Letter · Decision Letter 0]

11 Mar 2022

PONE-D-21-23455Association between epidemiological and clinico-pathological features of breast cancer with prognosis, family history, Ki-67 proliferation index and survival in Tunisian breast cancer patientsPLOS ONE

Dear Dr. Hamdi,

Thank you for submitting your manuscript to PLOS ONE. After careful consideration, we feel that it has merit but does not fully meet PLOS ONE’s publication criteria as it currently stands. Therefore, we invite you to submit a revised version of the manuscript that addresses the points raised during the review process. Please submit your revised manuscript by April 25, 2022. If you will need more time than this to complete your revisions, please reply to this message or contact the journal office at plosone@plos.org. Please include the following items when submitting your revised manuscript:A rebuttal letter that responds to each point raised by the academic editor and reviewer(s). You should upload this letter as a separate file labeled 'Response to Reviewers'.A marked-up copy of your manuscript that highlights changes made to the original version. You should upload this as a separate file labeled 'Revised Manuscript with Track Changes'.An unmarked version of your revised paper without tracked changes. You should upload this as a separate file labeled 'Manuscript'.

We look forward to receiving your revised manuscript.

Kind regards,

Elda Tagliabue

Academic Editor

PLOS ONE

Journal Requirements:

2. Please provide additional details regarding participant consent. In the Methods section, please ensure that you have specified (1) whether consent was informed and (2) what type you obtained (for instance, written or verbal). If your study included minors, state whether you obtained consent from parents or guardians. If the need for consent was waived by the ethics committee, please include this information.

Reviewers' comments:

Reviewer's Responses to Questions

**Comments to the Author**

1. Is the manuscript technically sound, and do the data support the conclusions?

Reviewer #1: Yes

Reviewer #2: Yes

2. Has the statistical analysis been performed appropriately and rigorously? 

Reviewer #1: No

Reviewer #2: Yes

3. Have the authors made all data underlying the findings in their manuscript fully available?

Reviewer #1: Yes

Reviewer #2: Yes

4. Is the manuscript presented in an intelligible fashion and written in standard English?

Reviewer #1: No

Reviewer #2: Yes

5. Review Comments to the Author

Reviewer #1: The article entitled “Association between epidemiological and clinico-pathological features of breast cancer with prognosis, family history, Ki-67 proliferation index and survival in Tunisian breast cancer patients” tackles an important problem in the Tunisian population.

However, some improvements are required before its publication.

1) Could you please deepen the description about the Medical Oncology Department at Abderrahmen Mami Hospital? Is it an important centre for all the country? Just part of it?

2) Could you specify the symptoms you used to define diagnostic delay? Are they self-reported by patients or are they derived from clinical examinations?

3) Could you write again how you performed the selection of familial cases? The criteria are not clear.

4) Could you write again the Statistical Analysis paragraph? You mixed indices to describe variables distribution and statistical test.

For continuous variables it is recommended to include also standard deviation, not just the mean.

5) In the Epidemiological Features paragraph you wrote “most prevalent comorbidities associated with breast cancer were…”, which other comorbidities have you considered? You should specify that you considered some comorbidities known to be associated with breast cancer from the literature, but you did not check in your study that association.

6) You should include in tables 1,2, and 3 the tests used to obtain each pvalue (Chi-Square or Fisher Exact or T-test). Pvalues lower than 0.001 should be reported as “<0.001”. Furthermore, it is not clear why there are three pvalues for T stage in table 1.

7) Table 3 reported the association between Ki-67 proliferation index and clinicopathological features of breast cancer, why did you include early age at onset? Could describe it in the main text?

8) I would suggest investigating your own ki-67 cut off predicting survival in overall luminal breast cancer tumours and compare it with those highlighted in the literature.

9) Table 4 is not clear. To which test are the pvalues referred?

Reviewer #2: the authors analyzed the epidemiological and clinico-pathological features of breast cancer in Tunisia and to determine prognostic factors with special interest to family history, Ki-67 proliferation index and comorbidity.

in this paper, the authors showed new insights into epidemiological and clinico-pathological characteristics of breast cancer in Tunisian population.

the study is well done.

6. PLOS authors have the option to publish the peer review history of their article (what does this mean?). If published, this will include your full peer review and any attached files.

Reviewer #1: No

Reviewer #2: No

---

## [Author Response · Author response to Decision Letter 0]

5 May 2022

PONE-D-21-23455

Association between epidemiological and clinico-pathological features of breast cancer with prognosis, family history, Ki-67 proliferation index and survival in Tunisian breast cancer patients. 

PLOS ONE 

Authors’ Responses: 

Thank you for giving us the opportunity to submit a revised version of our research paper. We have revised the manuscript according to the journal requirements and the reviewers’ comments and suggestions. All page and line numbers have been included; please refer to the revised manuscript file with tracked changes.

Journal Requirements:

Q1.Please ensure that your manuscript meets PLOS ONE's style requirements, including those for file naming. The PLOS ONE style templates can be found at 

A1: Thank you for this comment. As recommended, we have revised our manuscript to ensure that it meets all PLOS ONE’s style requirements and the following modifications have been made:

File naming: 

We have made modification on the name of the following supporting file: 

“S1 File” to “S1_File”.

Manuscript body formatting:

We have made modification on the style of the title “Survival in non-metastatic breast cancer patients” (18pt font.

Figures titles:

We have made modification on the figure’s title; we don’t use italics type.

Authors:

We added “Farouk Benna” as an author.

Affiliations:

Afficliations’s numbers have been updated after adding the affiliation of Farouk Benna. 

3 Department of Radiation Oncology, University of Tunis, Tunis, Tunisia.

4Laboratory of Human and Experimental Pathology, Institut Pasteur de Tunis, Tunis, Tunisia.

Additional modifications: 

Page 3, line65: we added “breast cancer patients from”.

Q2. Please provide additional details regarding participant consent. In the Methods section, please ensure that you have specified (1) whether consent was informed and (2) what type you obtained (for instance, written or verbal). If your study included minors, state whether you obtained consent from parents or guardians. If the need for consent was waived by the ethics committee, please include this information.

A2: We thank the reviewer for this relevant comment. Indeed, our study does not include minors. Our Institutional Review Board has approved our study which is a retrospective study, so the need for consent was waived by the ethics committee while maintaining patients´anonymity. As suggested, we provided more details regarding participant consent and ethics approval of our study. 

Page 4, lines 94- 95: “The need for consent has been waived by the ethics committee. However, personal data protection has been respected by preserving patient anonymity”.

Q3. We note that the grant information you provided in the ‘Funding Information’ and ‘Financial Disclosure’ sections do not match. 

A3: We thank the reviewer to pointing this out. Please consider grant information provided in the ‘Financial Disclosure’ section: “This study was supported by the Tunisian Ministry of Health (PEC-4-TUN) and the Tunisian Ministry of Higher Education and Scientific Research (LR16IPT05 and LR20IPT05)”.

We added information regarding the role of funders; “The funders had no role in study design, data collection and analysis, decision to publish, or preparation of the manuscript”.

As recommended, this has been updated in the ‘Funding Information’ section to match information provided within the ‘Financial Disclosure’ section.

Q4. In your Data Availability statement, you have not specified where the minimal data set underlying the results described in your manuscript can be found. PLOS defines a study's minimal data set as the underlying data used to reach the conclusions drawn in the manuscript and any additional data required to replicate the reported study findings in their entirety. All PLOS journals require that the minimal data set be made fully available. For more information about our data policy, please see http://journals.plos.org/plosone/s/data-availability.

A4: Thank you for raising this point. We confirm that all data underlying the findings described on our manuscript are fully available in the main text or provided as supporting information files.

Responses to Reviewers’ comments: 

Reviewer #1: The article entitled “Association between epidemiological and clinico-pathological features of breast cancer with prognosis, family history, Ki-67 proliferation index and survival in Tunisian breast cancer patients” tackles an important problem in the Tunisian population.

However, some improvements are required before its publication.

Q1) Could you please deepen the description about the Medical Oncology Department at Abderrahmen Mami Hospital? Is it an important centre for all the country? Just part of it?

A1: We would like to thank the reviewer for the careful reading of our manuscript and for the constructive comments. As recommended, we have provided more details about the Medical Oncology Department of Abderrahmen Mami Hospital. 

Page 4, lines 89- 92: ‘’ We retrospectively reviewed a cohort of 602 patients with histologically confirmed breast cancer treated in the period between 2011 and 2015, in Medical Oncology Department of “Abderrahmen Mami” Hospital, Tunisia, which is the second national reference center for chemotherapy and radiation in Tunisia, with dedicated multidisciplinary teams. Data were collected from patients’ medical record”.

Q2) Could you specify the symptoms you used to define diagnostic delay? Are they self-reportedby patients or are they derived from clinical examinations?

A2: Thank you for pointing this out. Delay in diagnosis is referring to the time between self-reported symptoms and final histological diagnosis of breast cancer and it does not include initiation of therapy. As suggested, we added more clarifications on this point in the method section: Page 4, line 98: “time between symptoms self-reported by patients and final histological diagnosis of breast cancer “.

Q3) Could you write again how you performed the selection of familial cases? The criteria are not clear.

A3: Thank you for your comment. As recommended, we reviewed the selection criteria of familial breast cancer cases to be more clear. Page 5, lines 111- 119: The selection of familial cases was based on several criteria mainly the family history of breast and ovarian cancers and the age at diagnosis; patients were selected if at least one of the following criteria was fulfilled: (1) The patient was diagnosed with breast cancer before the age of 36 years, (2) The patient was diagnosed with triple negative breast cancer regardless of age, (3) The patient has at least two first or second-degree relatives with breast cancer, (4) The patient has at least two first or second-degree relatives with breast or ovarian cancer regardless of age, and at least one case of pancreatic or prostate cancer”.

Q4) Could you write again the Statistical Analysis paragraph? You mixed indices to describe variables distribution and statistical test.

For continuous variables it is recommended to include also standard deviation, not just the mean.

A4: Thank you for pointing this out. As suggested, we reviewed the statistical analysis paragraph to be more comprehensive and clear (Page5, lines 125-141: ‘’Statistical analyses were performed using the Statistical Package for Social Sciences (SPSS Inc., Chicago, IL, USA) Version 23.0. Data were summarized by numbers and percentages for categorical variables, mean or median and range for continuous variables. Logistic regression was performed to evaluate prognostic factors.

The assessment of the association between familial breast cancer status, young age at onset, Ki-67 cut-off levels and prognostic factors was performed using Khi-2 statistical test. An independent t-test was used to compare the means of two groups.

We evaluated the prognostic value of several cut-off levels of Ki-67 in terms of overall survival (OS). We also considered different subgroups according to axillary lymph node involvement: pN0, 1-3pN+ and ≥4pN+. Survival curves were dressed according to the Kaplan-Meier analysis and compared with the log-rank test. .All p-values were two-sided, and p<0.05 was used to indicate a statistically significant difference’’.

As suggested, for continuous variables, we added standard deviation values.

Q5) In the Epidemiological Features paragraph you wrote “most prevalent comorbidities associated with breast cancer were…”, which other comorbidities have you considered? You should specify that you considered some comorbidities known to be associated with breast cancer from the literature, but you did not check in your study that association.

A5: Thank you for raising this point. Indeed, based on the literature review, we evaluated the presence of the most prevalent comorbidities known to be associated with breast cancer in our studied breast cancer cohort.

Based on the reviewer’s suggestion, we updated information regarding comorbidities in all over the manuscript: 

Abstract section, page2, line 35: “observed in the studied cohort” instead of “associated with breast cancer”.

Results section, page13, lines 255-256: “The most prevalent comorbidities observed in our breast cancer cohort were cardiovascular diseases (50.78%) and diabetes (18.32%)”.

Discussion section, Page17, line 366: We deleted “and comorbidities”.

Q6) You should include in tables 1,2, and 3 the tests used to obtain each pvalue (Chi-Square or Fisher Exact or T-test). Pvalues lower than 0.001 should be reported as “<0.001”.Furthermore, it is not clear why there are three pvalues for T stage in table 1.

A6: Thank you for this comment. We have made the suggested modifications. We provided more details regarding used statistical tests as a footnote in each table.

In addition, P values lower than 0.001 were reported as “<0.001” all over the manuscript.

In our study, we have compared epidemiological and clinico-pathological parameters between familial and sporadic breast cancer patients. For T stage variable, since there are 3 subgroups (T1-T2, T3 and T4), we have conducted Khi2 statistical test for each subgroup separately to determine in which one there are statistical difference between sporadic and familial cases. However, based on the reviewer’s comment, we updated the statistical test to be more coherent and homogeneous with the other studied parameters (Table1).

And we updated, information in table 1 as well as in the results section, page7, lines 196-197: “a significantly high tumor stage was noted among familial cases (p=0.048) (Table1)”.

Q7) Table 3 reported the association between Ki-67 proliferation index and clinicopathological features of breast cancer, why did you include early age at onset? Could describe it in the main text?

A7: Thank you for this comment. Indeed, the Tunisian population was characterized by a mean age at diagnosis 

younger than that reported in Western countries and by a relatively high incidence rate among young patients under 35 years. Furthermore, it is well known that young breast cancer patients have a poorer prognosis with high fraction of Ki-67 index. However, in Tunisia, the association between Ki-67 and early age at onset was not evaluated which prompt us to study the association between both parameters. We added further statement with reference in the discussion section, page15, lines 308-310: “In accordance with our results, other studies found that a higher Ki-67 index is significantly correlated with positive lymph node, an increased risk of recurrence, a high grading and an early age at onset ‘’.

Q8) I would suggest investigating your own ki-67 cut off predicting survival in overall luminal breast cancer tumours and compare it with those highlighted in the literature.

A8: We thank the reviewer for this comment. In fact, based on the literature review, the most debatable Ki-67 cut-offs predicting survival for luminal breast cancer cases are 14% and 20. In the current report, we explored these cut-offs and the 30% and 50% cut-offs and we observed a significant difference in OS with the Ki-67 cut-off of 30% in luminal breast cancer group.

Q9) Table 4 is not clear. To which test are the pvalues referred?

A9: For the study of Ki-67 cut-offs predicting survival in overall luminal breast cancer tumours (Table 4), we used Kaplan-Meir Method and the p values were derived from Log-Rank test. Information is provided in the method section, page 5, line 139: “Survival curves were dressed according to the Kaplan-Meier analysis and compared with the log-rank test”. We further noted this statement as a footnote in table 4. 

Reviewer #2: 

The authors analyzed the epidemiological and clinico-pathological features of breast cancer in Tunisia and to determine prognostic factors with special interest to family history, Ki-67 proliferation index and comorbidity.

In this paper, the authors showed new insights into epidemiological and clinico-pathological characteristics of breast cancer in Tunisian population.

The study is well done.

A1: We would like to thank the reviewer for the time and efforts dedicated to review our manuscript.

---

## [Decision Letter · Decision Letter 1]

27 May 2022

Association between epidemiological and clinico-pathological features of breast cancer with prognosis, family history, Ki-67 proliferation index and survival in Tunisian breast cancer patients

PONE-D-21-23455R1

Dear Dr. Hamdi,

We’re pleased to inform you that your manuscript has been judged scientifically suitable for publication and will be formally accepted for publication once it meets all outstanding technical requirements.

Kind regards,

Elda Tagliabue

Academic Editor

PLOS ONE

Additional Editor Comments (optional):

Reviewers' comments:

Reviewer's Responses to Questions

**Comments to the Author**

1. If the authors have adequately addressed your comments raised in a previous round of review and you feel that this manuscript is now acceptable for publication, you may indicate that here to bypass the “Comments to the Author” section, enter your conflict of interest statement in the “Confidential to Editor” section, and submit your "Accept" recommendation.

Reviewer #1: All comments have been addressed

2. Is the manuscript technically sound, and do the data support the conclusions?

Reviewer #1: Yes

3. Has the statistical analysis been performed appropriately and rigorously? 

Reviewer #1: Yes

4. Have the authors made all data underlying the findings in their manuscript fully available?

Reviewer #1: Yes

5. Is the manuscript presented in an intelligible fashion and written in standard English?

Reviewer #1: Yes

6. Review Comments to the Author

Reviewer #1: I would like to thank the authors for having improved the manuscript.

I do not have any additional comments.

7. PLOS authors have the option to publish the peer review history of their article (what does this mean?). If published, this will include your full peer review and any attached files.

Reviewer #1: No

---

## [Editor Report · Acceptance letter]

2 Sep 2022

PONE-D-21-23455R1 

Association between epidemiological and clinico-pathological features of breast cancer with prognosis, family history, Ki-67 proliferation index and survival in Tunisian breast cancer patients 

Dear Dr. Hamdi:

I'm pleased to inform you that your manuscript has been deemed suitable for publication in PLOS ONE. Congratulations! Your manuscript is now with our production department. 

Kind regards, 

on behalf of

Dr. Elda Tagliabue 

Academic Editor

PLOS ONE